# Non-rigid Point Cloud Registration with Neural Deformation Pyramid

**Yang Li**[1]
liyang@mi.t.u-tokyo.ac.jp

**Tatsuya Harada**[1,2]
harada@mi.t.u-tokyo.ac.jp

[1]The University of Tokyo

[2]RIKEN

## Abstract

Non-rigid point cloud registration is a key component in many computer vision and computer graphics applications. The high complexity of the unknown non-rigid motion make this task a challenging problem. In this paper, we break down this problem via hierarchical motion decomposition. Our method called Neural Deformation Pyramid (NDP) represents non-rigid motion using a pyramid architecture. Each pyramid level, denoted by a Multi-Layer Perception (MLP), takes as input a sinusoidally encoded 3D point and outputs its motion increments from the previous level. The sinusoidal function starts with a low input frequency and gradually increases when the pyramid level goes down. This allows a multi-level rigid to non-rigid motion decomposition and also speeds up the solving by 50 times compared to the existing MLP-based approach. Our method achieves advanced partial-to-partial non-rigid point cloud registration results on the 4DMatch/4DLoMatch benchmark under both no-learned and supervised settings. Code is available at https://github.com/rabbityl/DeformationPyramid.

Non-rigid point cloud registration is a key component in many computer vision and computer graphics applications. The goal of non-rigid registration is to find the transformation that maps one point cloud to another. With the availability of consumer range sensors that can measure time-varying surface points, non-rigid registration has been applied to dynamic shape reconstruction problems such as human performance capture, enabling a wide range of applications in XR and robotics.

Non-rigid registration is a challenging problem. First, 3D sensor measurements often contain noise, outliers, and occlusions. Occlusions often lead to disconnection of point cloud geometry. Point clouds may also have very low overlap ratios with each other due to the scene deformation and sensor's viewpoint change. The most challenging thing is, unlike rigid registration that only needs to determine the rotation and translation parameters, non-rigid registration needs to estimate the unknown movement of all points, making this problem especially complex to solve.

In this paper, we alleviate this complexity through motion decomposition. We observe that natural non-rigid motion usually forms a hierarchical structure: with higher hierarchies representing the global movements and lower hierarchies representing the local deformation. For instance, a walking person can be roughly approximated by three levels: 1) global location and orientation change, 2) local articulated movements from arms, legs, etc, and 3) fine-grained cloth deformation caused by exterior forces. Each level represents motion at a different scale and they have top-down dependencies.

Based on this observation, we propose a hierarchical motion representation called Neural Deformation Pyramid (NDP) for non-rigid registration. NDP has a pyramid architecture. Each pyramid level contains a Multi-Layer Perception (MLP) that takes as input a sinusoidally encoded 3D point and outputs its motion increments from the previous level. We found that the frequency of the sinusoidal function controls MLP's capacity of representing non-rigidity: low frequencies yield smooth signals

36th Conference on Neural Information Processing Systems (NeurIPS 2022).

that are suitable for fitting relatively rigid motion; high frequencies produce more fluctuations that are capable of representing highly non-rigid motion. We start the sinusoidal function at the first pyramid level with a low frequency and gradually increase it when the pyramid level goes down. This allows a multi-level rigid to non-rigid motion decomposition and also achieves over 50 times faster solving than the existing MLP-based approach.

The paper is about developing an MLP-based hierarchical deformation model for non-rigid point cloud registration. The proposed method achieves state-of-the-art partial-to-partial non-rigid registration results on the challenging 4DMatch/4DLoMatch [19] benchmark under both no-learned and supervised settings. We also demonstrate the application for shape transfer.

## 1 Related Work

**Non-rigid point cloud registration.** Non-rigid point cloud registration is about estimating the deformation field or assigning point-to-point mapping from one point cloud to another. The simplest way is to estimate the point-wise parameters, such as affine transform, under motion smoothness regularization [22]. Optimal transport-based registration method [11] finds displacement for each point using a global bijective-matching constraint. Coherent Point Drift (CPD) [27, 14] constructs a 3D displacement field using Gaussian Mixture Model (GMM) and encourages coherent motion of nearby points. Deformation graph [34] represents the scene using a sparsely sub-sampled graph from the surface and propagates deformation from node to surface via "skinning". The Non-rigid Iterative Closest Point (NICP) [16] achieve efficient registration by optimizing the alignment energy and deformation graph regularization cost [33], and has been adopted in many real-time 4D reconstruction systems [28]. Li et al. [18] and Bozic et al. [4] learn dense feature alignment or correspondence re-weighting by differentiating through deformation graph-based non-rigid optimization. Learning-based scene flow estimation methods, e.g. FlowNet3D [8], use 3D siamese networks to regress the 3D displacement field between two point clouds and reach real-time inference but are not robust under large deformations and ambiguities. Lepard [19] use Transformer [37] to learn global point-to-point mapping and use it as landmark to guide global non-rigid registration. Functional map approaches [29] estimate the correspondence in the spectral domain between computed Laplacian-Beltrami basis functions, but usually assume connected surfaces thus not suitable for real-world partial point cloud data. To alleviate this problem, Synorim [15] uses 3D CNN networks to estimate the basis functions and derive scene flow from the functional mapping. While existing methods mainly represent non-rigid motion or mapping at a single level, this paper propose a multi-level deformation model for non-rigid point cloud registration.

**Pyramid for motion estimation.** Bouguet et al. [3] show that the pyramid implementation of the Lucas-Kanade [5] method improves feature tracking. PWCNet [35] estimates optical flow on a pyramid of CNN feature maps. PointPWC-net [38] adopts similar idea for scene flow estimation from point cloud inputs. Pyramid-based camera 6-Dof pose estimation can be found in SLAM systems [9]. ZoomOut [23] adopts upsampling in the spectral domain for shape correspondence estimation. DynamicFusion [28] constructs a tree of deformation graph for non-rigid tracking, and shows that it stabilizes tracking and reduces computation cost. DeepCap [12] combines an inner body pose with an outer cloth deformation for human performance capture. This paper is about using MLP to create a hierarchical deformation pyramid for non-rigid point cloud registration of general scenes.

**Motion field with coordinate-MLP.** Coordinate-MLP uses an MLP to map input coordinates to signal values. Coordinate-MLP is continuous and memory-efficient. It has shown promising results for representing 1D sound wave [32], 2D images [32], 3D shape [24], and radiance field [25], etc. Tancik et al. [36] show that tuning the sinusoidal positional encoding of the input coordinate bias the network to fitting low- or high-frequency signals. Coordinate-MLP is also suitable for representing the deformation field. Close to our work, Li et al. [21] and Li et al. [17] use MLP to estimate scene flow, Nerfies [30] and SAPE [13] propose a coarse-to-fine motion field MLP optimization technique by progressively expanding the frequency bandwidth of the input positional encoding. However, the aforementioned motion field MLPs are black box models that are designed to represent signals at a single scale, and usually need a large network to fit complex motion, as a result their optimizations are usually time-consuming. This paper decomposes the motion field using a sequence of smaller MLPs, achieving more interpretable and controllable motion representation and faster optimization.

## 2 Non-rigid Point Cloud Registration Notation

Given a source point cloud $\mathbf{S} = \{\mathbf{x}_i | \mathbf{x}_i \in \mathbb{R}^3, i = 1, ..., n_1\}$ and a target point cloud $\mathbf{T} = \{\mathbf{y}_j | \mathbf{y}_j \in \mathbb{R}^3, j = 1, ..., n_2\}$, where $n_1, n_2$ are the number of points, our goal is to recover the non-rigid warp function $\mathcal{W} : \mathbb{R}^3 \mapsto \mathbb{R}^3$ that transforms points from $\mathbf{S}$ to $\mathbf{T}$. The simplest form of the warp function is a dense $\mathbb{R}^3$ vector field, which is also known as scene flow. Scene flow is in theory sufficient to represent any continuous deformation, but in practice, it can not fit non-linear motions very well, such as 3D rotations. We therefore formulate the non-rigid warp function using the $\text{SE}(3)$ field.

**Dense** $\text{SE}(3)$ **warp field**. Given a globally non-rigidly deforming point cloud, we consider each individual point locally undergoes 3D rigid body movement. A 3D rigid body transform $\begin{pmatrix} \mathbf{R} & \mathbf{t} \\ 0 & 1 \end{pmatrix} \in \text{SE}(3)$ denotes rotation and translation in 3D, with $\mathbf{R} \in \text{SO}(3)$ and $\mathbf{t} \in \mathbb{R}^3$. We parameterize rotations with a 3-dimensional axis-angle vector $\boldsymbol{\omega} \in \mathbb{R}^3$. We use the exponential map $\exp : \mathfrak{so}(3) \to \text{SO}(3)$, $\widehat{\boldsymbol{\omega}} \mapsto e^{\widehat{\boldsymbol{\omega}}} = \mathbf{R}$ to convert from axis-angle to matrix rotation form, where the $\widehat{\cdot}$-operator creates a $3 \times 3$ skew-symmetric matrix from a 3-dimensional vector. The resulting 3D motion parameterization for a point $\mathbf{x}_i \in \mathbb{R}^3$ is therefore denoted by $\boldsymbol{\xi}_i = (\boldsymbol{\omega}_i, \mathbf{t}_i) \in \mathbb{R}^6$, i.e. each point has 6 degrees of freedom (Dof). The warp function reads

$$\mathcal{W}(\mathbf{x}_i, \boldsymbol{\xi}_i) = e^{\widehat{\boldsymbol{\omega}}_i} \mathbf{x}_i + \mathbf{t}_i \tag{1}$$

**Dense** $\text{Sim}(3)$ **warp field**. This paper mainly focuses on registering point clouds captured from the same scene with the same scale. For scale variant tasks such as inter-shape registration (c.f. Sec. 4.3), we extend the above warp function by incorporating a scaling factor $s \in \mathbb{R}^+$, resulting in a 3D similarity transform $\begin{pmatrix} s\mathbf{R} & \mathbf{t} \\ 0 & 1 \end{pmatrix} \in \text{Sim}(3)$. With the parameterization $\boldsymbol{\xi}_i = (s_i, \boldsymbol{\omega}_i, \mathbf{t}_i) \in \mathbb{R}^7$, the warp function reads $\mathcal{W}(\mathbf{x}_i, \boldsymbol{\xi}_i) = s_i e^{\widehat{\boldsymbol{\omega}}_i} \mathbf{x}_i + \mathbf{t}_i$.

We shows ablation study of the dense warp field types including $\mathbb{R}^3$, $\text{SE}(3)$, and $\text{Sim}(3)$, and rotations representations including Axis-angle, Euler-angle, Quanterion, and 6D [40] in Sec. 4.2.

## 3 Neural Deformation Pyramid

As denoted Sec. 2, we aim to find the motion parameter $\boldsymbol{\xi}_i$ for each 3D point $\mathbf{x}_i$ in the source point cloud. However, due to the high complexity and non-convexity of the non-rigid registration, directly estimating $\boldsymbol{\xi}_i$ is usually difficult and time-consuming. Therefore, we want to decompose $\boldsymbol{\xi}_i$ into a sequence of sub-transformations $\{\boldsymbol{\xi}_i^1, \boldsymbol{\xi}_i^2, ..., \boldsymbol{\xi}_i^m\}$, such that each sub-transformation is easier to estimate, and when combining them we can get the same registration effect as $\boldsymbol{\xi}_i$.

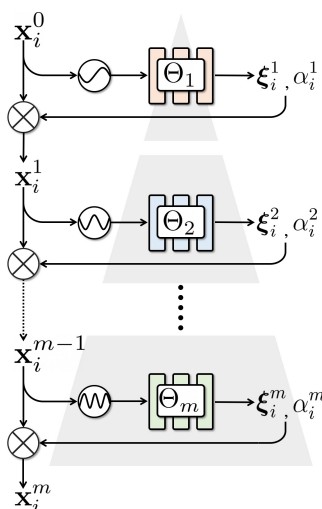

Figure 1: Neural Deformation Pyramid.

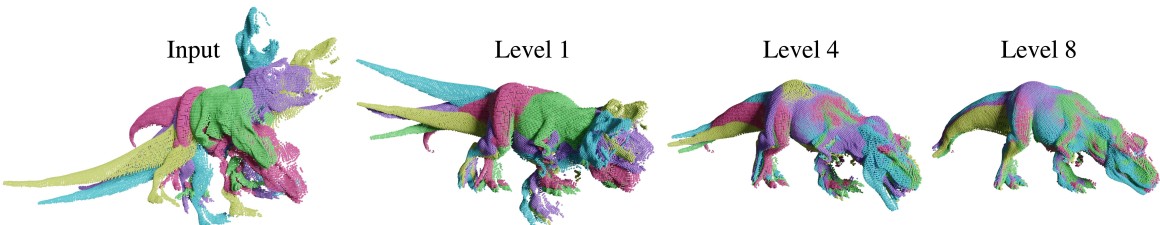

| Input | Level 1 | Level 4 | Level 8 |

Figure 2: Hierarchical non-rigid registration of multiple scans of a Dinosaur using LNDP. We show the outputs from the 1st, 4th, and 8th level of the deformation pyramid. The pink color denotes the target point cloud, the rests are the sources (the figures in this paper are best viewed on screen).

To this end, we introduce Neural Deformation Pyramid (NDP) which allows hierarchical motion decomposition using a pyramid architecture.

## 3.1 Hierarchical motion decomposition.

As shown in Fig. 3, NDP is a pyramid of functions $\Delta = \{(\Gamma_k, \Theta_k) | k = 1, .., m\}$, where each pyramid level contains a pair of continuous functions $(\Gamma_k, \Theta_k)$, and $m$ is the total number of levels.

At level $k$ of the pyramid, $\Gamma_k$ is the positional encoding function that maps the input point from previous level $\mathbf{x}_i^{k-1} \in \mathbb{R}^3$ using the sinusoidal encoding

$$\Gamma_k : \mathbb{R}^3 \mapsto \mathbb{R}^6, \quad \mathbf{x}_i^{k-1} \mapsto \Gamma_k(\mathbf{x}_i^{k-1}) = (\sin(2^{k+k_0}\mathbf{x}_i^{k-1}), \cos(2^{k+k_0}\mathbf{x}_i^{k-1})) \tag{2}$$

where $k_0$ is a constant that controls the initial frequency at the first level of the pyramid. The frequency of the sinusoidal function is enssential for motion decomposition: low frequencies yield smooth signals that are suitable for fitting relatively rigid motion; high frequencies produce more fluctuations that are capable of representing highly non-rigid motion. By gradually increasing frequency with pyramid level $k$, we can achieve a hierarchical rigid-to-nonrigid motion decomposition. Fig. 2 shows an example of hierarchical registration of of multiple scans of a Dinosaur.

$\Theta_k$ is an optimizable MLP network. It takes as input the encoded coordinate $\Gamma_k(\mathbf{x}_i^{k-1})$ and estimates the transformation increments at the current pyramid level. Formally, we have

$$\Theta_k : \mathbb{R}^6 \mapsto \mathbb{R}^7, \quad \Gamma_k(\mathbf{x}_i^{k-1}) \mapsto \Theta_k(\Gamma_k(\mathbf{x}_i^{k-1})) = (\boldsymbol{\xi}_i^k, \alpha_i^k) \tag{3}$$

where $\boldsymbol{\xi}_i^k \in \mathbb{R}^6$ is the 6-Dof transformation parameter that is obtained from a liner output head of the MLP. $\alpha_i^k \in [0, 1]$ is a scalar that represents the network's confidence in if the motion estimates are successful at the current level, it is obtained via a Sigmoid output head of the MLP.

Given the 6-Dof estimates $\boldsymbol{\xi}_i^k$ and confidence $\alpha_i^k$, we compose the transformed coordinate at the $k$-th level by

$$\mathbf{x}_i^k \leftarrow \mathbf{x}_i^{k-1} + \alpha_i^k \cdot \mathcal{W}(\mathbf{x}_i^{k-1}, \boldsymbol{\xi}_i^k) \tag{4}$$

where $\mathcal{W}$ is the warp function as defined in Eqn. 1. Note that $\alpha_i^k$ control the degree of deviation w.r.t to the previous more rigid pyramid level, thus we regard it as the level-wise *deformability*. To encourage as-rigid-as-possible movement, we apply regularization terms on $\alpha_i^k$ which is shown in the next.

**Merits of hierarchical motion decomposition.** 1) it provides a more controllable and interpretable coordinate MLP-based deformation representation. 2) it simplifies the task and speedup the optimization: we can use a small MLP for each level, because each level only needs to estimate the motion increments at a single frequency band, we also found that the overall convergence of the pyramid is faster than using a single coordinate-MLP with the full frequency band, this significantly cut off the total optimization time compared to the existing MLP-based approaches, c.f. discussion in Sec. 4.2.

## 3.2 Cost function.

At level $k$, we denote the transformed source point cloud as $\mathbf{S}^k = \{\mathbf{x}_i^k | i = 1, ..., n_1\}$. The target point cloud is $\mathbf{T} = \{\mathbf{y}_j | j = 1, ..., n_2\}$. The cost functions at level $k$ are:

**Chamfer distance term.** Chamfer distance finds the nearest point in the other point cloud, and sums the square of distance. Formally it is defined as

$$E_{cd}^k = \frac{1}{|\mathbf{S}^k|} \sum_{\mathbf{x}_i^k \in \mathbf{S}^k} \min_{\mathbf{y}_j \in \mathbf{T}} \rho(\mathbf{x}_i^k - \mathbf{y}_j) + \frac{1}{|\mathbf{T}|} \sum_{\mathbf{y}_j \in \mathbf{T}} \min_{\mathbf{x}_i^k \in \mathbf{S}^k} \rho(\mathbf{x}_i^k - \mathbf{y}_j) \tag{5}$$

where $\rho(.,.)$ is the distance function, for which we use $L_1$ or $L_2$ norm. We found that the robust $L_1$ norm is more suitable for handling partial-to-partial registration, see ablation study in Section. 4.

**Correspondence term.** Given a putative correspondence set $\mathcal{M}$, we minimize

$$E_{cor}^k = \frac{1}{|\mathcal{M}|} \sum_{(u,v) \in \mathcal{M}} \rho(\mathbf{x}_u^k - \mathbf{y}_v) \tag{6}$$

where $(u, v) \in \mathcal{M}$ are the indices of the matched points in $\mathbf{S}^k$ and $\mathbf{T}$. To obtain correspondence, we leverage the learning-based point cloud matching method Lepard [19], which predicts sparse point-to-point matches. Lepard's prediction contains a certain amount of outlier correspondences, which may lead to erroneous registration. To alleviate this, we design a Transformer-based outlier rejection method, which takes as input matched coordinates pairs $(u, v) \in \mathbb{R}^6$ and estimates their outlier probabilities. The details can be seen in the supplemental material.

**Deformability regularization term.** Given the deformability score $\alpha_i^k$ in Eqn. 4, we encourages zero predictions by minimizing the negative log-likelihood

$$E_{reg}^k = \frac{1}{|\mathbf{S}^k|} \sum_{\mathbf{x}_i \in \mathbf{S}^k} -\log(1 - \alpha_i^k) \tag{7}$$

This is to encourage as-rigid-as-possible movement. We found this regularization help preserve the geometry of the point cloud given high frequency input signals.

**Total cost function.** The total cost function $E_{total}^k$ combines the above terms with the weighting factors $\lambda_{cd}$, $\lambda_{cor}$, and $\lambda_{reg}$ to balance them :

$$E_{total}^k = \lambda_{cd} E_{cd}^k + \lambda_{cor} E_{cor}^k + \lambda_{reg} E_{reg}^k \tag{8}$$

In the case that we use the correspondence term $E_{cor}$, we denote our method as LNDP where 'L' indicates the Learned correspondences.

### 3.3 Non-rigid registration algorithm

Non-rigid registration using NDP is performed in a top-down way: the top-level MLP is firstly optimized, once it converges, proceeds to a lower level, we repeat this till all MLPs are optimized. We use gradient descent as the optimizer. Optimization of an MLP stops if 1) the $max\_iter = 500$ is reached, 2) a given registration cost threshold $\gamma = 0.0001$ is reached, or 3) the registration cost does not change for more than $\sigma = 15$ iterations. Alg. 1 shows the pseudocode of the algorithm.

---

**Algorithm 1** Non-rigid registration using NDP

---

1: **function** REGISTRATION($\mathbf{S}^0, \mathbf{T}$)                   ▷ $\mathbf{S}^0$ denotes the raw source point cloud
2:     **for** $k \leftarrow 1$ to $m$ **do**
3:         $\Theta_k \leftarrow$ XavierUniform()                   ▷ MLP initialization
4:         **for** $iter \leftarrow 1$ to $max\_iter$ **do**
5:             $\mathbf{S}^k \leftarrow$ TransformSourcePointCloud($\mathbf{S}^{k-1}, \Gamma_k, \Theta_k$)
6:             $E_{total}^k \leftarrow$ ComputeRegistrationCost($\mathbf{S}^k, \mathbf{T}$)
7:             **if** ConvergenceConditionSatisfied($E_{total}^k, iter$) **then**        ▷ Early stop
8:                 **break**
9:             $\Theta_k \leftarrow$ GradientDecentSolver($\Theta_k, E_{total}^k$)        ▷ Update MLP weights
10:     **return** $\mathbf{S}^m$

---

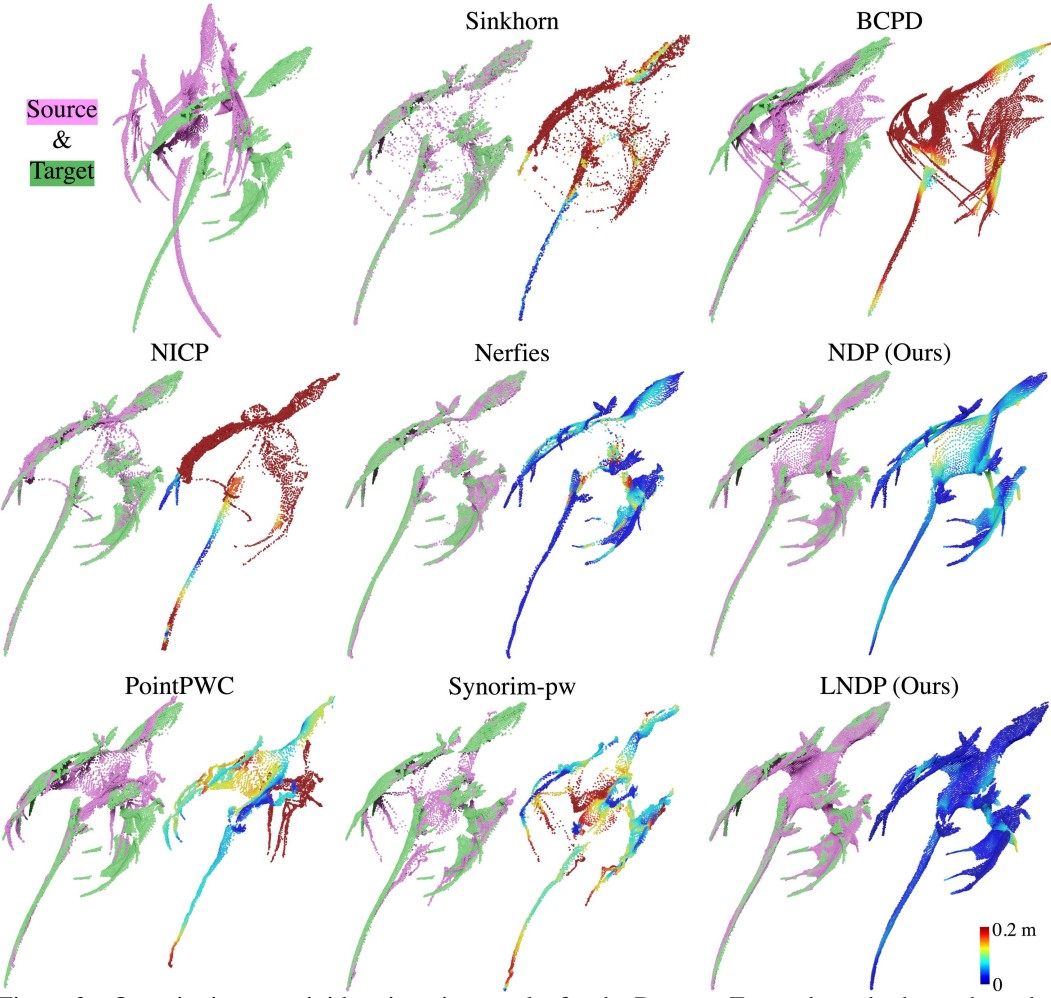

Figure 3: Quantitative non-rigid registration results for the Dragon. For each method, we show the point cloud alignment and error map. Sinkhorn does not preserve the point cloud topology. CPD and NICP can easily fall into local minima. The scene flow produced by PointPWC and Synonym-pw distorts the geometry. Synonym-pw does not generalize well to uncommon shapes in the dataset such as this dragon.

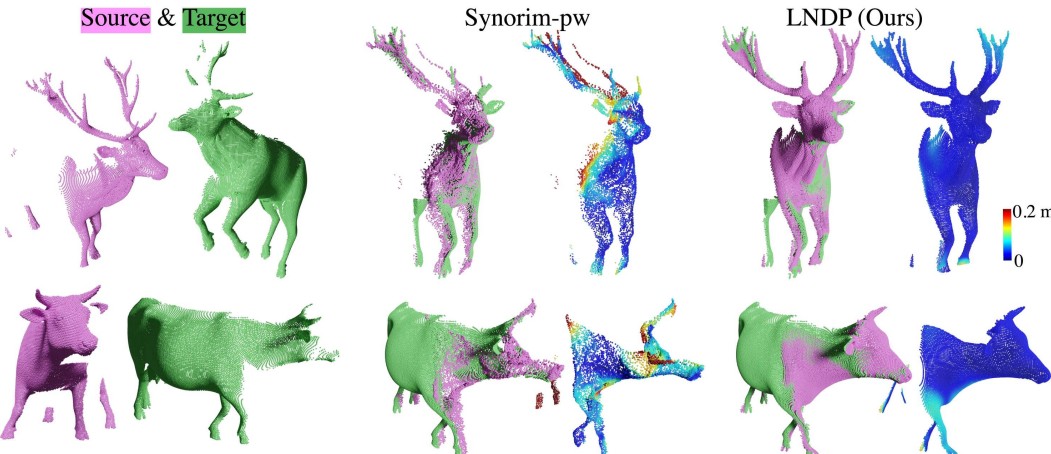

Figure 4: Quantitative non-rigid registration results for the Deer and Cow. Synorim-pw runs on subsampled 8192 points. LNDP can directly warp all input points because it is a continuous function. LNDP better preserves the geometry, especially for the non-overlapping regions.

# 4 Experiments

## 4.1 Benchmarking partial-to-partial non-rigid point cloud registration

**4DMatch/4DLoMatch benchamrk.** 4DMatch/4DLoMatch [19] is a benchmark for non-rigid point cloud registration. It is constructed using animation sequences from DeformingThings4D [20]. This benchmark is extremely challenging due to the partial overlap, occlusions, and large motion present in the data. Point cloud pairs in this benchmark have a wide range of overlap ratios: $45\% - 92\%$ in 4DMatch and $15\% - 45\%$ in 4DLoMatch. We found that the original benchmark contains a certain amount of examples that are dominated by rigid movement. For fair evaluation, we remove data with near-rigid movements, please check the supplementary for details.

Table 1: Quantitative non-rigid registration results on 4DMatch and 4DLoMatch.

| | Method | 4DMatch | | | | 4DLoMatch | | | | |
| | | EPE↓ | AccS ↑ | AccR ↑ | Outlier↓ | EPE↓ | AccS↑ | AccR↑ | Outlier↓ | Time ↓ |
|---|---|---|---|---|---|---|---|---|---|---|
| No-Learned | ICP [2]† | 0.296 | 2.96 | 12.06 | 71.50 | 0.565 | 0.14 | 0.74 | 90.87 | 0.10 |
| | ZoomOut [23] | 0.598 | 1.82 | 4.23 | 89.27 | 0.663 | 0.22 | 0.81 | 90.39 | 151.10 |
| | CPD [27] | 0.274 | 1.57 | 7.30 | 74.52 | **0.463** | 0.07 | 0.48 | 84.49 | 4.52 |
| | BCPD [14] | 0.291 | 5.13 | 12.35 | 73.74 | 0.492 | 0.20 | 0.86 | 86.88 | 5.81 |
| | Sinkhorn [11] | 0.308 | 2.76 | 8.13 | 79.86 | 0.505 | 0.20 | 0.81 | 89.47 | 3.76 |
| | NICP [28] | 0.325 | 6.44 | 12.10 | 80.04 | 0.517 | 0.18 | 0.73 | 92.37 | 4.80 ‡ |
| | NSFP [17] | 0.265 | 8.66 | 18.65 | 64.96 | 0.495 | 0.38 | 1.56 | 84.77 | 39.54 ‡ |
| | Nerfies [30] | 0.280 | 12.65 | 25.41 | 58.91 | 0.498 | **1.05** | 3.01 | 82.21 | 115.94 ‡ |
| | NDP (Ours) | **0.195** | **18.69** | **35.64** | **45.04** | **0.467** | 0.79 | **3.05** | **80.47** | 2.31 ‡ |
| Supervised | Lepard [19]+SVD† | 0.137 | 6.91 | 24.50 | 43.43 | 0.160 | 5.27 | 19.77 | 44.16 | 0.06 ‡ |
| | PointPWC [38] | 0.182 | 6.25 | 21.49 | 52.07 | 0.279 | 1.69 | 8.15 | 55.70 | 0.06 ‡ |
| | FLOT [31] | 0.133 | 7.66 | 27.15 | 40.49 | 0.210 | 2.73 | 13.08 | 42.51 | 0.07 ‡ |
| | GeomFmaps [6] | 0.152 | 12.34 | 32.56 | 37.90 | 0.148 | 1.85 | 6.51 | 64.63 | 135.18 |
| | Synorim-pw [15] | 0.099 | 22.91 | 49.86 | 26.01 | 0.170 | 10.55 | 30.17 | **31.12** | 0.41‡ |
| | Lepard+NICP [19] | 0.097 | 51.93 | 65.32 | 23.02 | 0.283 | 16.80 | 26.39 | 52.99 | 3.93 ‡ |
| | LNDP (Ours) | **0.075** | **62.85** | **75.26** | **16.78** | **0.169** | **28.65** | **43.37** | 32.14 | 2.39 ‡ |

† rigid registration methods.    ‡ with GPU accerleration (NVIDIA A100).

**Metrics.** The metrics for evaluating non-rigid registration quality are 1) End-Point Error (EPE), i.e., the average norm of the 3D warp error vectors over all points, 2) 3D Accuracy Strict (AccS), the percentage of points whose relative error < 2.5% or < 2.5 cm, 3) 3D Accuracy Relaxed (AccR), the percentage of points whose relative error < 5% or < 5 cm, and 4) Outlier Ratio, the percentage of points whose relative error > 30%. We consider AccS and AccR as the most important metrics as they exactly measure the ratio of accurately registered points.

**Baselines.** We benchmark non-rigid registration using a number of baselines under both the no-learned and supervised settings. We use the underlined names for brevity:

- **No-Learned.** Point-to-point Iterative Closest Point (ICP) [2] implemented in Open3D [39], Coherent Point Drift (CPD)[27] and its Bayesian formulation BCPD [14], ZoomOut [23], Sinkhorn optimal transport method implemented in Geomloss [11] and Keops [10], point-to-point Non-rigid ICP (NICP) [28], coordinate-MLP based approaches including Neural Scene Flow Prior (NSFP) [17] and Nerfies [30].

- **Supervised.** Procrustes approach using Lepard [19]'s feature matching followed by SVD solver [1] (Lepard+SVD), Deep Geometric Maps (GeomFmaps) [7], Synorim [15] in the pair-wise setting (Synorim-pw), scene flow estimation methods including FLOT [31] and PointPWC [38], feature matching enhanced NICP as in [19] (Lepard+NICP). All supervised models are re-trained on 4DMatch's training split before evaluation.

**Benchmarking results.** Tab. 1 shows the quantitative non-rigid registration results. The coordinate-MLP-based methods, including NSFP, Nerfies, and our NDP get clearly better results than other no-learned baselines. This indicates the advantages of using the continuous coordinate-MLP to represent motion. A major drawback of Nerfies and NSFP is that their optimization is usually time-consuming. With hierarchical motion decomposition, our NDP runs around 50 times faster than Nerfies and still be equally or more accurate. Note that on 4DMatch, NDP even outperforms the supervised methods FLOT and PointPWC on the AccS/AccR metrics by a significant margin. On 4DLoMatch, due to the small point cloud overlap ($15\% - 45\%$), none of the no-learned methods can produce a reasonable result. LNDP obtains significantly better non-rigid registration results than other supervised baselines. Fig. 2, 3, and 4 shows the qualitative non-rigid registration results.

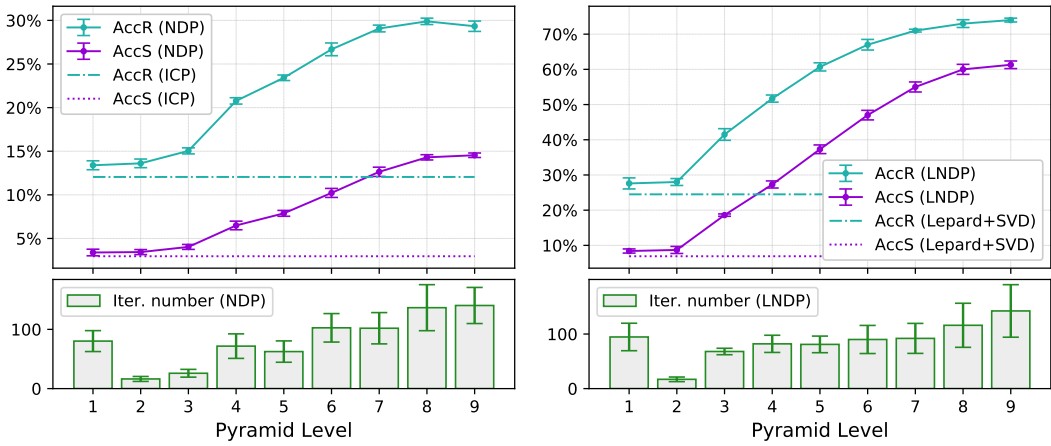

Figure 5: Point cloud registration accuracies (top) and gradient descent iterations (bottom) at each pyramid level on the 4DMatch benchmark. Left: NDP, right: LNDP.

Table 2: Ablation study of loss functions.

| | 4DMatch | | | | 4DLoMatch | | | |
|---|---|---|---|---|---|---|---|---|
| Method | EPE↓ | AccS ↑ | AccR ↑ | Outlier↓ | EPE↓ | AccS↑ | AccR↑ | Outlier↓ |
| NDP ($L_2$ Chamfer distance term ) | 0.205 | 14.34 | 29.79 | 48.4 | 0.473 | 0.82 | 3.03 | 80.92 |
| NDP ($L_1$ Chamfer distance term ) | **0.195** | **18.69** | **35.64** | **45.04** | **0.467** | **0.79** | **3.05** | **80.47** |
| LNDP ($L_2$ Correspondence term ) | 0.078 | 61.27 | 74.10 | 17.50 | 0.177 | 26.59 | 41.50 | 33.81 |
| LNDP ($L_1$ Correspondence term ) | **0.075** | **62.85** | **75.26** | **16.78** | **0.169** | **28.65** | **43.37** | **32.14** |

Table 3: Ablation study of warp field type and rotation representation in LNDP.

| | | 4DMatch | | | | 4DLoMatch | | | | |
|---|---|---|---|---|---|---|---|---|---|---|
| Warp field | Rotation format | EPE↓ | AccS↑ | AccR↑ | Outliers↓ | EPE↓ | AccS↑ | AccR↑ | Outliers↓ | Iter.↓ |
| $\mathbb{R}^3$ | – | 0.084 | 58.03 | 70.91 | 19.15 | 0.197 | 21.46 | 34.30 | 40.00 | 882 |
| | 6D [40] | 0.080 | 56.15 | 72.04 | 18.71 | 0.172 | 26.95 | 43.09 | 32.14 | 1077 |
| SE(3) | Quaternion | 0.080 | 55.47 | 71.87 | 18.73 | **0.168** | 26.79 | 43.29 | **31.33** | 1010 |
| | Euler angle | **0.075** | 62.72 | 75.21 | 16.80 | 0.169 | **28.89** | **43.68** | 32.17 | 799 |
| | Axis-angle (Default) | **0.075** | **62.85** | **75.26** | **16.78** | 0.169 | 28.65 | 43.37 | 32.14 | **785** |
| Sim(3) | Axis-angle | 0.079 | 59.33 | 72.35 | 19.86 | 0.173 | 24.74 | 42.56 | 33.01 | 1388 |

## 4.2 Ablation study

**Pyramid level.** We test 9 pyramid levels with the initial frequency parameters in Eqn. 2 set to $k_0 = -8$. Fig. 5 shows that, 1) the first level only slightly surpasses the rigid registration baseline, 2) the registration accuracies gradually increase with the pyramid level, i.e., the capacity of representing non-rigid motion gradually grows, and 3) lower levels tend to need more iterations to converge.

**Why is NDP faster than Nerfies?** Summing up the iterations from all levels (c.f. Fig. 5), NDP needs a total of 738 gradient decent iterations to converge. The coarse-to-fine optimization in Nerfies uses a single MLP with the full frequency band as input. As a comparison, Nerfies needs 3792 iterations for convergence. This indicates that our motion decomposition method speeds up the overall convergence. In addition, to represent complex motions, Nerfies needs a large MLP of $(\text{witdth}, \text{depth}) = (128, 7)$. As a comparison, for each pyramid level we use a small MLP of $(\text{witdth}, \text{depth}) = (128, 3)$, because each level only needs to estimate the motion increments at a single frequency band. Note that NDP only queries a single level MLP at an iteration (c.f. Algorihtm. 1). As a result, the overall computation overhead of NDP is much smaller than Nerfies.

**$L_1$ norm vs $L_2$ norm for partial registration.** We show in Table. 2 that the $L_1$ norm is more suitable for partial-to-partial registration, because it is more robust to large errors. For example, the chamber distance term is based on the sum of the squared distance from each point to the other model. Minimizing this loss would attempt to reduce the distance for every point, including those that do not correspond to any point on the other point cloud due to the partial overlap. Compared to the $L_2$ norm, the $L_1$ norm allow for large distances on some points. Similarly, the $L_1$ norm is also more tolerant to outliers in the correspondence term.

**Motion field type.** Tab. 3 shows that, 1) $\mathrm{SE}(3)$ warp field gets better results than $\mathrm{Sim}(3)$ field and $\mathbb{R}^3$ vector field, and 2) $\mathrm{Sim}(3)$ field requires the most iterations to converge, possibly because it needs to estimate the extra scale factor.

**Rotation representation.** Tab. 3 shows that Axis-angle and Euler angle get similar results and are better and converge faster than Quanterion and 6D.

## 4.3 Scale variant registration with Sim(3) warp field

In Fig. 6, we provide examples of the "shape transfer" as an application of our non-rigid registration method. To handle scale change between different shapes, we employ the dense $\mathrm{Sim}(3)$ warp field (c.f. Sec. 2). The dense formulation allows NDP to reflect different scale changes at different regions of the body, e.g. character **A**'s stomach is expanding while the legs are shrinking. The original mesh vertices of the shapes are unevenly distributed, which is not suitable for computing the chamfer distance cost as in Eqn. 5, therefore we use uniformly sub-sampled point clouds from the mesh's surface as input points. Since NDP is a continuous function, the deformed mesh can be obtained by querying NDP using the original mesh's vertices. We use the registration parameters $(m, k_0) = (9, -8)$, this allows the transformed shapes roughly match the target shapes while retaining the geometrical details of the source shapes.

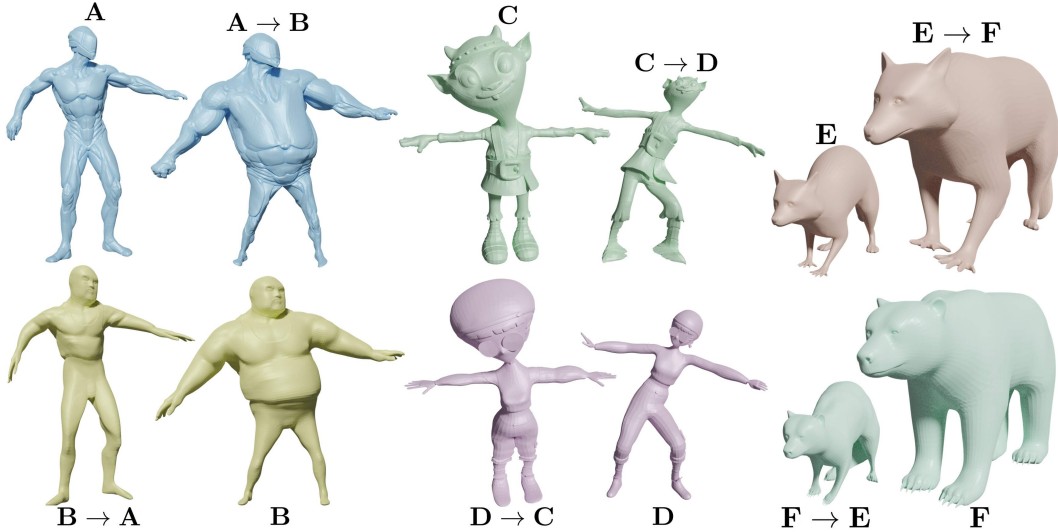

Figure 6: Shape transfer using NDP. The input shapes **A**, **B**, **C**, and **D** are "Alien Soldier", "Ortiz", "Doozy", and "Jackie" from Mixamo (https://www.mixamo.com/); **E** and **F** are "Racoon" and "Bear" from DeformingThings4D [20]. The arrows indicate the directions of transfer.

# 5   Conclusion

We show that non-rigid point cloud registration can be decomposed into a hierarchical motion estimation schema by stacking coordinate networks with growing input frequency. Our method demonstrates superior non-rigid registration results on the 4DMatch partial-to-partial non-rigid registration benchmark under both no-learned and supervised settings. Our method runs over 50 times faster than the existing coordinate-networks-based approach.

**Limitations.**   1) NDP uses input coordinates that are defined over the 3D Euclidean space, therefore it can not handle topological changes very well, extending NDP to the manifold surface space would be interesting. 2) Common to the unsupervised approach, NDP does not handle non-isometric cases. A failure case could be seen in Figure. 6: i.e. A's chest is mapped to B's belly. 3) Though NDP runs about 50 times faster than Nerfies, the current implementation still does not run at a real-time rate, further speedup could leverage the tiny CUDA neural network framework [26]. 4) Finally, non-rigid registration in the low-overlap cases, such as examples in 4DLoMatch, is still challenging, our method cannot solve all the cases.

**Broader Impact.**   Our paper presents non-rigid registration, which is needed for a variety of applications ranging from XR to robotics. In the former, a precise modeling of dynamic and deformable objects is of major importance to provide an immersive experience to the user. In the later, localizing and understanding dynamic objects such as humans and animals in the enviroment using 3D sensor is essential for safe and intelligent robot operation. On the other hand, as a low-level building block, our work has no direct negative outcome, other than what could arise from the aforementioned applications.

## Acknowledgments and Disclosure of Funding

This work was partially supported by JST AIP Acceleration Research JPMJCR20U3, Moonshot R&D Grant Number JPMJPS2011, CREST Grant Number JPMJCR2015, JSPS KAKENHI Grant Number JP19H01115 and Basic Research Grant (Super AI) of Institute for AI and Beyond of the University of Tokyo.

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
