# OpenReview forum: "Non-rigid Point Cloud Registration with Neural Deformation Pyramid"
_NeurIPS.cc/2022/Conference — NeurIPS 2022 Accept_

### Official Review · Reviewer_iTyU · 2022-07-05

**Rating:** 7
**Confidence:** 4
**Soundness:** 3 good
**Presentation:** 3 good
**Contribution:** 3 good

**Summary:**

The paper proposes a non-rigid registration algorithm that also works in the partial-to-partial point clouds setting. The idea is to regularize the deformation in three ways: 1) by dividing the problem into pyramidal (hierarchical) sub-problems; 2) by considering first low-frequency deformations, and scaling to higher frequencies later; 3) by defining for each point a rigid (or affine) deformation (i.e., rotation, translation, and optional scaling). The method is tested on 4DMatch and 4DLoMatch, showing compelling results and outperforming several SOTA methods.


**Questions:**

- How much the method is sensitive to the introduction of noise, clutter, and point-cloud density?
- How does the method computation timing scale at a different number of points?

**Limitations:**

Both societal impact and limitations are fairly discussed.

**Strengths And Weaknesses:**

== STRENGHTS ==
- The method seems novel in the composition of its elements. I like the proposed regularizations on the different aspects of the problem. The approach seems relatively straightforward, while probably to be implementable by scratch, further details are required (e.g., a deeper explanation of the architecture implementation, fewer pointers to different papers for details)
- The shown results are convincing on a significant variety of shapes. The attached video gives a nice insight into the low-to-high frequencies deformation.

== WEAKNESSES ==
- The method is tested on various shapes, but the quantitative results are only on near-isometric deformations of synthetic data. The only experiment between significantly different shapes (Figure 6) is not convincing in terms of structural preservation of the shapes (the left-most example shows that A's chest is mapped to B's belly).
- No analysis of noise or clutter is provided. This makes unclear the applicability of the method in real-case scenarios. Also, it mentions the possibility of including the scale factor, but no quantitative results are provided in this case. No analysis of different amounts of point clouds partiality or computational scaling at the different number of points.

Minor fixes:
- line 125: Fig.3 -> Fig. 1
- line 137: 'registration of of multiple scans'
- line 168: 'this regularization help to preserve' -> helps
- [14] is not introduced in the previous works; [a] would be a better alternative than [6] since it overcome some LBO limitations in the point coluds setting

[a]: Correspondence learning via linearly-invariant embedding, Marin et al., 2020

== AFTER REBUTTAL ==

After the rebuttal and the discussion phase, I am prone to keep my initial rating and vote for acceptance. For the final version, I suggest including in the main manuscript the feedback about the method limitations.

---

> ### Author Response · Authors · 2022-07-28
> **Response to Reviewer iTyU**
>
> Thank you for your review and your questions!
>
>
> ###  1. Experiments on non-isometric deformations, cluttered scene, and point density change,  in real-world data.
> We evaluate our method on two **real-world** benchmarks: DeepDeform, and KITTI Scene Flow.
> The results are posted in the 2nd paragraph on the [Response to Reviewer xwkn (part 1/2)](https://openreview.net/forum?id=pfEIGgDstz0&noteId=HUvm3_PYhH8).
> Our NDP produces advanced results on both benchmarks.
> In particular, the Lidar scans in KITTI capture running vehicles, walking pedestrians, and static trees/buildings on the road, i.e., the data is cluttered, and breaks the isometric deformation assumption. The point density of Lidar scans also changes drastically from near to far.
> NDP is robust to the above factors and produces competitive results.
>
>
>
> &nbsp;
> ### 2. Registration result at different level of noise.
> The following table shows the registration accuracy of NDP under different ratios of point cloud noise.
> A noisy point is created via uniform perturbation of a clean point inside a ball with a radius=0.5m (the size of objects in 4DMatch range from 0.6m to 2.1m). We do not observe a significant performance drop until the introduction of 25% noise, while real-world range sensors such as the Kinect1 camera only produces 4%~6% noise, see [Tanwi et al., 2014]. This experiment, combined with the results on DeepDeform and KITTI,  proves that our method is robust to noisy data in real-world scenarios.
>
> - **Registration Accuracy Relaxed (AccR) of NDP on 4DMatch under different level of noise**
> |     Noise ratio                  |  &ensp; 0% &ensp;  |  &ensp; 5% &ensp; |  &ensp;10%&ensp;  |  &ensp;15% &ensp; | &ensp; 20% &ensp; |  &ensp;25%&ensp;  |  &ensp;30% &ensp; |  &ensp;35% &ensp; |  &ensp;40% &ensp; |&ensp;  45%&ensp;  |  &ensp;50%&ensp;  |
> |-|:-----:|:-----:|:-:|:-:|:-:|:-:|:-----:|:---:|:-----:|:-----:|:-----:|
> | AccR (%)  | 29.81 | 28.76 | 27.74 | 27.08 | 27.00 | 24.43 | 24.25 | 24.22 | 23.89 | 22.64 | 21.83 |
>
>
> [Tanwi et al., 2014] Characterizations of Noise in Kinect Depth Images: A Review. IEEE Sensor 2014.
>
> &nbsp;
> ### 3. Computation time at a different number of points. ###
> To conclude first, **the time complexity is sub-linear or $\mathcal{O}(1)$ with the number of points.**
> The dominant computation overhead is from the network optimization, the time complexity is $\mathcal{O} (n\times m) $, where $n$ is the number of points and $m$ is the number of iterations required for convergence.
> If $m$ is fixed, the time grows in $\mathcal{O} (n)$ of the number of points.
> However, we found that when increasing training point $n$,  the total number of iteration $m$ decreases.
> Therefore, if we use all points for optimization, the registration time grows sub-linearly.
> For faster registration, we can optimize only sub-sampled points. This keeps a constant registration time regardless of the input size. The following table proves this argument.
>
> - **Registration time (s) at different number of input points**
> | Complexity|   |  2k  |  4k  |  6k  |   8k  |  10k  |  12k  |  14k  |  16k  |  18k  |  20k  |
> |--|---|:----:|:-:|:-:|:-:|:-:|:-:|:-:|:-:|:-:|:-:|
> | $\mathcal{O}(n)$ |  | 2.72 | 5.42 | 8.13 | 10.84 | 13.55 | 16.25 | 18.97 | 21.68 | 24.39 | 27.10 |
> | sub-linear       | NDP (use all points for optimization)      | 2.72 | 3.56 | 3.79 |  3.78 |  4.84 |  5.06 |  5.07 |  5.70 |  6.38 |  5.91 |
> | $\mathcal{O}(1)$ | NDP (sub-sample 2k points for optimization) | 2.72 | 2.58 | 2.60 |  2.57 |  2.47 |  2.82 |  2.63 |  2.76 |  2.58 |  2.87 |
>
>
>
> &nbsp;
> ### 4. Registration result at different level of point clouds partiality.
> In pairwise setting, we use overlap ratio $\theta$ to denote the partiality.
> The results are posted in the 3nd paragraph on the [Response to Reviewer xwkn (part 1/2)](https://openreview.net/forum?id=pfEIGgDstz0&noteId=HUvm3_PYhH8).
>
>
>
> &nbsp;
> ###  5. "A's chest is mapped to B's belly"
> This is indeed a drawback of NDP: it is trying to found correspondence in nearest regions, which may fail under large deformation.
> But please note that this is the result of an unsupervised method, such artifacts could be resolved with sparse landmark correspondence, e.g. using sparse chest-to-chest, belly-to-belly links as complementary training signal as in the LNDP case. We leave this as future work.
>
>
> &nbsp;
> ### 6. Others
> - "further details are required for implementation": we add more implementation details at issue 7 in [Response to Reviewer xwkn (part 2/2) ](https://openreview.net/forum?id=pfEIGgDstz0&noteId=oG-dtzjSYOp), we will also publish the source code.
> - "quantitative results for scale registration": we are planning to create such a registration benchmark with scale changes, and report results in a revised version.
> - Thanks, we fixed the typos.
> - We added [14] in related work.
> - We cited and discussed [Marin et al., 2020] and will try to evaluate this method in a revised version.

---

> > ### Comment · Reviewer_iTyU · 2022-08-07
> > **Rebuttal answer**
> >
> > I thank the authors for their effort to address my concerns.
> >
> > I have carefully read the rebuttal and other reviewers' comments. In my original review, my main concerns were about different challenging scenarios to test the robustness of the method. I see that further experiments have been provided, and I am satisfied with these additions, while the authors agree that there are limitations about significantly non-isometric cases common to the unsupervised approach. I suggest including this observation in the main manuscript limitations.
> >
> > I see that other reviewers, in general, agree that the idea is sound and interesting, while maybe not dramatically innovative from a general perspective. Looking forward to discussing this with other reviewers!

---

### Official Review · Reviewer_xwkn · 2022-07-07

**Rating:** 4
**Confidence:** 3
**Soundness:** 2 fair
**Presentation:** 3 good
**Contribution:** 2 fair

**Summary:**

This paper proposes a novel method - Neural Deformation Pyramid (NDP) for non-rigid point cloud registration. Non-rigid point cloud registration remains a challenging problem as the input 3D data often contains noise, outliers, and occlusions. The proposed method NDP deals with the above complexity by decomposing motion hierarchically. The NDP method uses coordinate-MLP as the basic structure, similar to existing works Nerfies [30] and SAPE [13]. The NDP outperforms existing methods on the 4DMatch/4DLoMatch [19] dataset and is faster than existing MLP-based methods.


**Questions:**

- The authors claim that the proposed method is faster than the existing method; however, there is no detailed data about the runtime speed of the proposed method and existing methods. The author should report the speed in detail in their paper.
- In line 53, what does 'via skinning’ mean here?
- There are not enough implementation details to reproduce the proposed method, such as the parameters of the MLP structures.
- In 3.3 (line 174), the authors propose the non-rigid registration algorithm. However, there is not enough justification for the algorithm. Such as why the authors optimize different levels in order instead of updating all levels simultaneously. Also, there is no justification why the max_iter is set to 500 and the theta to 15 iterations.


**Limitations:**

The authors have adequately addressed the limitations and potential negative societal impact.


**Strengths And Weaknesses:**

## Strengths

- The proposed method NDP is well-motivated, and the connection to existing methods is clearly stated.
- The proposed method achieves better performance on both no-learned and supervised settings quantitatively and qualitatively than previous methods and is faster than the existing method Nerfies [30].
- The authors compared their proposed methods with a variety of existing methods.
- This paper is well written and easy to follow.


## Weaknesses
- One of the paper's main contributions is introducing a pyramid architecture to the non-rigid motion registration task. However, there is a range of methods that have already discussed pyramid structures, as justified in Related Work (line 68 to 76). Therefore, there is a lack of technical contribution in this paper. Similarly, the proposed framework is based on coordinate-MLP structure, similar to existing methods Nerfies [30] and SAPE [13].
- The authors only conducted experiments on a single benchmark 4DMatch/4DLoMatch [19]. The authors should consider conducting experiments on other benchmarks, such as the 3DMatch/3DLoMatch used in previous research [19].
- In line 25, the authors claim that the proposed method alleviates the complexity of non-rigid registration task (data noise, outliers, and occlusions). However, there are no quantitative experimental results regarding the above complexities. The authors should consider conducting experiments with the proposed method to data noise, outliers, and occlusions.

---

> ### Author Response · Authors · 2022-07-29
> **Response to Reviewer xwkn (part 2/2)**
>
>
>
> &nbsp;
> ### 5. "there is no detailed data about the runtime"
> Actually, we have reported the runtime of all baseline methods, please check the right-most column of Table. 1.  Specifically, the  runtime (s) of **Nerfies** vs **Ours** is **115.94** vs **2.31**, i.e. Ours runs 50 times faster.
>
> &nbsp;
> ### 6. "In line 53, what does 'via skinning’ mean here?"
> Skinning is the idea of transforming a point cloud by a
> blend of multiple transformations. A standard algorithm is Linear Blend Skinning (LBS) which uses weighted linear combination of the transformations. We will add explanations.
>
> &nbsp;
> ### 7. "not enough implementation details to reproduce the method, such as the MLP structures."
> The MLP structure is $(width, depth)=(128,3)$ for all pyramid levels, see line 229.
> We also show the pseudocode and other hyper-parameters in Sec. 3.3.
> We will clarify other configs as in the following table.
> Note that, as a part of the submission, we have included the source code and data link, which will be made publicly available.
>  | &ensp;pyramid level&ensp; | &ensp;learning rate &ensp;| &ensp;activation &ensp;|&ensp; weight decay &ensp;|&ensp; momentum &ensp;|
> |:-------------:|:-------------:|:----------:|:------------:|:--------:|
>    |       9       |      0.01     |   Relu()   |    0.0001    |    0.9   |
>
>
> &nbsp;
> ### 8. "why the authors optimize different levels in order instead of updating all levels simultaneously?"
> There are two reasons: 1) **natural non-rigid motion has a top-down dependency**: the fine level motion is built on the coarse level motion, and it appears more intuitive to estimate the fine level motion when the coarse level motion is known. 2)  **updating MLPs at all levels simultaneously is time-consuming**: it requires $m$ times more computations than updating a single MLP, where $m$ is the number of levels in the pyramid.
> Refer to line.25-30 and line. 223-231 for more discussion.
>
>
> &nbsp;
> ### 9. "Why max_iter=500 and $\sigma$=15 ?"
> On average, the MLP in a pyramid level takes around 70~150 iterations to converge, see Figure 5. Therefore, we consider a maximum of 500 iterations as a safe upper bound for optimization. By observing the optimization loss curve, we found that the loss does not go down anymore if the loss has already stay unchanged for the past 15 iterations, therefore we set $\sigma=15$ as the convergence criteria. We will clarify.

---

> ### Author Response · Authors · 2022-07-30
> **Response to Reviewer xwkn (part 1/2)**
>
> Thank you for your review and your questions! Below we discuss the concerns you raised in the review and answer your questions.
> &nbsp;
> ### 1. Contribution.
>
> The contribution of this paper is the introduction of the first neural deformation pyramid model for non-rigid point cloud registration. Existing MLP-based approaches such as Nerfies and SAPE **are black boxes models** that only **represent motion signals at a single scale**, and usually **need a large network to fit complex motion**, as a result, their **optimization is time-consuming**.
> Our method has the following advantages compared to existing MLP-based approaches:
> - **Interpretability**. The multi-scale deformation representation makes the implicit neural network more interpretable.
> - **Task simplification**.  By decomposing the complicated non-rigid motion estimation task into a sequence of easier sub-tasks, we manage to simplify this problem. Results prove that this strategy yield more accurate registration than Nerfies.
> - **Optimization speedup**. By replacing the large MLP with a sequence of smaller MLPs, our method's optimization is 50 times faster than Nerfies.
>
>
> &nbsp;
>  ### 2. Experiments on other benchmarks.
> We add additional results on two real-world benchmarks: DeepDeform [Bozic et al.],  and KITTI Scene Flow [Geiger et al.].
> We omit 3DMatch/3DLoMatch because it is a rigid registration benchmark, while this paper focuses on non-rigid registration.
>
> -  **Registration results of no-learned methods on DeepDeform benchmark.**  (DeepDeform contains real-world partial RGB-D scans of dynamic objects, including humans, animals, cloth, etc. )
> |            | &ensp; EPE(cm) &ensp;  |&ensp;  AccS(%) &ensp; |&ensp;  AccR(%) &ensp; | &ensp; Outlier(%)&ensp;  |
> |:------------|:-------:|:-------:|:-------:|:----------:|
> | ZoomOut [23] |  2.88  | 62.31    | 85.74    |  19.55    |
> | Sinkhorn [11] |   4.08  |  42.49  |  77.41 |    23.85  |
> | NICP [28]  | 3.66   | 48.16    |  80.16 |     21.34  |
> | Nerfies [30] |   2.97  |  61.58  |  86.82  |    16.11  |
> | NDP (Ours) |   **2.13**  |  **79.01**  |  **94.09**  |    **11.55**   |
>
>
>
>
>
> - **Registration results on KITTI Scene Flow benchmark.** (KITTI contains Lidar scans in dynamic autonomous driving scenes. )
> |            | &ensp; Supervised  &ensp;| &ensp; EPE(m) &ensp;  |&ensp;  AccS(%) &ensp; |&ensp;  AccR(%) &ensp; |
> |:------------|:-------:|:-------:|:-------:|:-------:|
> | FlowNet3D [8]   | Yes|0.199        |    10.44     |   38.89  |
> | PointPWC [38]       |Yes|     0.142    |    29.91     | 59.83  |
> | NDP (Ours)      |   No|  **0.141**      |    **47.00**       | **71.20**   |
>
>
>
> DeepDeform [Bozic et al.]: DeepDeform: Learning Non-rigid RGB-D Reconstruction with Semi-supervised Data, CVPR 2020.
> KITTI [Geiger et al.]: Are we ready for autonomous driving? the KITTI vision benchmark suite, CVPR 2012.
>
>
> &nbsp;
> ### 3. Experiment with different level of Overlap/Partiality/Occlusion.
> We actually already conducted such experiments, please see the results on 4DMatch/4DLoMatch benchmark in Table 1.
>
> We want to clarify that, **in the pair wise setting, the three terms: overlap, partiality, and occlusion are connected**.
> Given $\text{Overlap ratio} =\theta$, we can obtain the others by $\text{Partiality}=\theta$, and $\text{Occlusion ratio}=(1-\theta)$.
> Because overlap ratio is defined by the the percentage of **co-visible** point between a pair of point clouds,
> by this definition, overlap ratio exactly represents the relative partiality between two point clouds.
> Occlusion ratio denotes the ratio of points that are **invisible** from another point cloud, therefore it is can be computed by $1-\theta$. The following table shows the stats of $\theta$ in 4DMatch/4DLoMatch. Note that our method is state-of-the-art on this benchmark.
> - **Statistics of Overlap/Partiality/Occlusion ratio in 4DMatch/4DLoMatch:**
> || &ensp;&ensp; Overlap ratio / Partiality ($\theta$ ) &ensp;&ensp; |&ensp; &ensp;Occlusion ratio ($1-\theta$)&ensp;&ensp;|
> |:-|:-:|:-:|
> |4DMatch| 45%~92% |8%~55%|
> |4DLoMatch| 15%~45% |55%~85%|
>
>
> We further re-group the results on 4DMatch/4DLoMatch based on different ranges of $\theta$ with 20% interval.
> - **Registration Accuracy (AccR) at different level of overlap ratio**
> | Overlap ratio ($\theta$)  | &ensp; < 20% &ensp; | &ensp; 20% ~ 40% &ensp;| &ensp; 40% ~ 60% &ensp; |  &ensp; 60% ~ 80%  &ensp;| &ensp; > 80% &ensp; |
> |-----------|:------:|:-------:|:-------:|:-------:|:--------:|
> | AccR (%) with NDP       |  0.97  |   2.80  |   8.16  |  25.57  |   63.65  |
> | AccR (%) with LNDP      |    **19.01**    |    **39.32**     |     **63.37**    |    **71.91**     |    **89.77**      |
>
>
>
> &nbsp;
>  ### 4. Experiments with noise/outliers.
> We have added experimental results and discussions in the 2nd paragraph at [Reply to Reviewer iTyU](https://openreview.net/forum?id=pfEIGgDstz0&noteId=gsKgJsUNiO8Z).

---

> > ### Comment · Reviewer_xwkn · 2022-08-07
> > **Rebuttal comment**
> >
> > The author feedback addresses most of my concerns. Taking into account the comments from other reviewers and the author feedback, I am inclined to accept this paper, but rejecting would it not be that bad.

---

### Official Review · Reviewer_AmBj · 2022-07-12

**Rating:** 4
**Confidence:** 4
**Soundness:** 2 fair
**Presentation:** 3 good
**Contribution:** 2 fair

**Summary:**

The paper proposes a neural model for non-rigid point cloud registration. The key idea is to decompose the desired non-rigid deformation into a hierarchy of rigid/similarity deformations, and train a neural model to predict the deformation at each level. The proposed method performs well in the presented evaluation.

The main contribution of this paper is the idea of hierarchical deformation and the proposed neural model to predict it.

**Questions:**

See the weaknesses above.

**Limitations:**

It is adequate.

**Strengths And Weaknesses:**

Strengths:
- The idea of hierarchical neural deformation is interesting.
- The method performs well in the evaluation.

Weaknesses:
- It is unclear why the proposed loss function is suitable for partial-to-partial registration. In particular, the chamber distance is based on sum of squared distance from each point to the other model. Minimizing this loss would attempt to reduce the distance for *every* point, including those that do not correspond to any point on the other model due to the partial overlap. In fact, existing work for robust non-rigid registration typically identifies such L2 distance as a source of poor performance for partial-to-partial registration and replaces it with other robust norms (e.g., L1) to allow for large distances on some points. It is surprising that using such a term in the training loss would work well for partial-to-partial registration in the current paper (for the unsupervised case at least).

---

> ### Author Response · Authors · 2022-07-31
> **Response to Reviewer AmBj**
>
> Thank you for the review and the helpful comments! Below, we answer your questions.
>
> ### Why the proposed loss function is suitable for partial-to-partial registration?
> In the supervised case,  partial-to-partial registration is made possible by relying on the correspondence term (in equation 6) as the dominant loss function.  The correspondence is obtained using the learning base point-to-point matching method Lepard [19]. We further use a transformer to reject match outliers. Finally, we get high-quality correspondences that are suitable for partial non-rigid registration.
>
> In the unsupervised case, the problem you mentioned does pose a challenge for registration: minimizing chamfer distance will push all points to move towards the target, and eventually, all source points may collapse on the target.
> Our technique that potentially mitigate this issue is the deformation regularization term (in Equation. 7): it assigns lower weights to the movements of the points that break the local rigidity assumption, i.e. preventing severe collapses of the point cloud geometry.  The down-weighted points will not be completely transported towards the target (c.f. Equation. 4), which allows partial alignment.
> This regularization term reduces the end-point-error, c.f. Table 2.
> Nevertheless, the unsupervised setting is still challenging especially for low overlap cases. It turns out that none of the unsupervised methods could produce reasonable results on 4DLoMatch (with overlap ratio < 45%), as already discussed in line. 214-216.
>
> We test both L1 and L2 norms of chamfer distance as loss terms. L1 norm does improve the metrics as shown in the following table. We will add this. However, please note that the focus of this paper is proposing a hierarchical neural deformation model for non-rigid registration, but not robust loss functions.
>
> - **Ablation study of loss function in unsupervised case on 4DMatch**
> |                       |  &ensp; EPE(cm)&ensp;  |  &ensp;AccS(%) &ensp; | &ensp; AccR(%) &ensp; | &ensp;Outlier(%) &ensp;|
> |-----------------------|:---------:|:---------:|:---------:|:----------:|
> | chamfer distance (L2 norm) |   20.05   |   14.34   |   29.79   |    48.4    |
> | chamfer distance (L1 norm) | **19.85** | **18.48** | **35.31** |  **45.64** |
> |                       |           |           |           |            |

---

### Official Review · Reviewer_hpiz · 2022-07-16

**Rating:** 7
**Confidence:** 4
**Soundness:** 3 good
**Presentation:** 4 excellent
**Contribution:** 3 good

**Summary:**

This paper presents Neural Deformation Pyramid (NDP), a learning-based method for non-rigid point cloud registration.
Given a source point cloud and a destination point cloud, NDP learns a dense SE(3) warp field (or dense Sim(3) warp field in the case of scaling difference), that is, to learn a rigid transformation for each point in the source cloud, so that the source cloud is best aligned with the destination cloud.

The proposed NDP seeks this warp filed in a hierarchical manner. Specifically, the warp field is decomposed into multiple layers of learnable MLP's, where at each layer the input 3D coordinates are passed through sinusoidal encoding with increasing frequencies. In this way, the NDP decomposes the non-rigid transformation into rigid transformations that go from a global scale to more local refinements. I found this design to be very intuitive.

Experiments compare NDP with a pretty large set of both non-learned methods and learned methods, and demonstrates the superior performance of NDP in terms of accuracy and efficiency.

**Questions:**

Is NDP (instead of LNDP) basically solving an optimization problem without any supervision?
The Chamfer distance and regularization term do not seem to require any groundtruth data.
If so, do you find the initialization of the network weights affect the performance of the optimization problem? (in terms of quality of registration)

**Ethics Review Area:**

["I don’t know"]

**Limitations:**

Limitations included.

**Strengths And Weaknesses:**

Strengths:
+ The intuition of hierarchical motion decomposition is intuitive and well motivated in the introduction.
+ The design of the network and its description is succinct and to the point.
+ Experiments are comprehensive, and support the claimed benefits of the algorithm.
+ Limitations are honestly acknowledged.

---

> ### Author Response · Authors · 2022-07-30
> **Response to Reviewer hpiz**
>
> Thanks for reviewing our paper! In the following, we answer your questions.
>
>
> ### 1. Is NDP (instead of LNDP) basically solving an optimization problem without any supervision?
> Yes, NDP is solving an online optimization problem without any supervision.
>
> ### 2. Does the initialization of the network weights affect the performance of the optimization problem?
> Yes, changing the initialization strategy could greatly affects the results of registration.
> The following table shows an ablation study for network initialization via Pytorch built-in functions.
> With *.ones_* initialization, the model does not converge at all.
> *.kaiming_uniform_* and *.xavier_uniform_* initialization produce similar results.
> We will add this.
>
>
> - **Ablation study of Pytorch init functions for NDP on 4DMatch:**
> |   *torch.nn.init*                       | EPE(cm) | AccS(%) | AccR(%) | Outlier(%) |
> |--------------------------|:-------:|:-------:|:-------:|:----------:|
> | *.ones_*                   |   47706.20  |   0.00   |  0.00    |  100.00  |
> | *.zeros_*                    |  21.83  |   3.48  |  13.89  |    60.00  |
> | *.kaiming_uniform_*          |  20.89    | **14.93**      | **29.81**     |  50.34       |
> | *.xavier_uniform_*  (Default) |  **20.05**  |  14.34  |  29.79  |   **48.4**    |
> |                  |     |      |      |         |
>
> In addition, we employ a trick to constraint the output of MLP: we apply a small scaling factor of $0.0001$  on the output of MLP, this encourages the MLP to produce a near-identity  $SE(3)$ matrix at the beginning of optimization.
> We found this trick is crucial for NDP (no-learned) but not necessary for LNDP (supervised). We will add discussions.

---

### Author Response · Authors · 2022-08-02
**General Reply**

 We would like to thank all reviewers for their detailed feedback!

We are glad that the reviews found our hierarchical neural deformation model is "intuitive"[hpiz], "interesting"[AmBj], "well-motivated" [xwkn,hpiz] and "novel"[iTyU];
the experiments are "comprehensive"[hpiz], includes results from "a pretty large set"[hpiz] or "a variety"[xwkn] of existing methods;
the proposed method "performs well in the evaluation"[AmBj], runs "faster"[xwkn], and "outperform several SOTA methods"[iTyU];
the  results are "convincing"[iTyU];
and the paper is also "well written and easy to follow"[xwkn].

We propose the first hierarchical neural deformation model for non-rigid point cloud registration.
Our method not only demonstrates superior non-rigid registration results on public benchmarks but also runs faster than existing coordinate-networks-based approaches.

To address some concerns of the reviewers, we conduct additional experiments including:
- Ablation study of network initialization strategies. [hpiz]
- Ablation study of loss functions. [AmBj]
- Evaluation on real-world benchmarks.[xwkn,iTyU]
- Experiments on different levels of noise, occlusion/partiality, input size, etc.[xwkn,iTyU]

We encourage the reviewers to also read our responses to the other reviewers.

Thank you,

the authors

---

### Meta-Review · Area_Chair_BeWp · 2022-08-24

**Recommendation:** Accept
**Confidence:** Certain

**Metareview:**

All reviewers agree this work is a creative approach to nonrigid registration, which is particularly hard in the mesh-free point cloud setting.  The discussion between the authors and reviewers was extremely productive and addressed most of the major concerns about this work.

In preparing the camera ready, the authors are encouraged to incorporate the new tables of results that appear in the discussion below into the paper and/or supplemental materials, to make sure it is archived.

**Award:**

No

---

### Decision · Program_Chairs · 2022-09-14

Accept